# An IoT Framework for Screening of COVID-19 Using Real-Time Data from Wearable Sensors

**DOI:** 10.3390/ijerph18084022

**Published:** 2021-04-12

**Authors:** Hamid Mukhtar, Saeed Rubaiee, Moez Krichen, Roobaea Alroobaea

**Affiliations:** 1Department of Computer Science, College of Computers and Information Technology, Taif University, Taif 21944, Saudi Arabia; r.robai@tu.edu.sa; 2Department of Industrial and Systems Engineering, College of Engineering, University of Jeddah, Jeddah 21577, Saudi Arabia; salrubaiee@uj.edu.sa; 3Department of Computer Science, Faculty of Computer Science and Information Technology, Al-Baha University, Al-Baha 65431, Saudi Arabia; moez.krichen@redcad.org; 4ReDCAD Laboratory, National School of Engineers of Sfax, University of Sfax, Sfax 3038, Tunisia

**Keywords:** coronavirus, IoT, Arduino, algorithm, cough, heartbeat

## Abstract

Experts have predicted that COVID-19 may prevail for many months or even years before it can be completely eliminated. A major problem in its cure is its early screening and detection, which will decide on its treatment. Due to the fast contactless spreading of the virus, its screening is unusually difficult. Moreover, the results of COVID-19 tests may take up to 48 h. That is enough time for the virus to worsen the health of the affected person. The health community needs effective means for identification of the virus in the shortest possible time. In this study, we invent a medical device utilized consisting of composable sensors to monitor remotely and in real-time the health status of those who have symptoms of the coronavirus or those infected with it. The device comprises wearable medical sensors integrated using the Arduino hardware interfacing and a smartphone application. An IoT framework is deployed at the backend through which various devices can communicate in real-time. The medical device is applied to determine the patient’s critical status of the effects of the coronavirus or its symptoms using heartbeat, cough, temperature and Oxygen concentration (SpO${}_{2}$) that are evaluated using our custom algorithm. Until now, it has been found that many coronavirus patients remain asymptomatic, but in case of known symptoms, a person can be quickly identified with our device. It also allows doctors to examine their patients without the need for physical direct contact with them to reduce the possibility of infection. Our solution uses rule-based decision-making based on the physiological data of a person obtained through sensors. These rules allow to classify a person as healthy or having a possibility of infection by the coronavirus. The advantage of using rules for patient’s classification is that the rules can be updated as new findings emerge from time to time. In this article, we explain the details of the sensors, the smartphone application, and the associated IoT framework for real-time, remote screening of COVID-19.

## 1. Introduction

Coronavirus is a large group of viruses that can be pathogenic in animals or humans. The novel coronavirus that was recently discovered is responsible for the coronavirus disease 2019 (COVID-19), which is a contagious illness caused by the last discovered type of coronavirus, SARS-CoV-2. This new virus and the disease were not known before the outbreak in Wuhan, China, in December 2019 [1]. According to the Centers for Disease Control and Prevention (CDC), at least seven different human coronaviruses are known so far and this number may increase in the future [2]. As of now, COVID-19 epidemic has resulted in about 2.6 million deaths as well as 117 million cases so far world-wide.

COVID-19 is transmitted through respiratory droplets expelled from the nose or mouth when a sick person coughs, sneezes, or talks. These drops are relatively heavy and do not cover large distances. Rather, they fall quickly on the ground [3]. COVID-19 can be contracted if these droplets are inhaled. For this reason, it is so important that we stay at least one meter away from others [4,5]. These droplets can be found on objects or surfaces (tables, door handles, ramps, etc.) around a sick person. COVID-19 can then be contracted if one touches these objects or surfaces and then touches their eyes, nose, or mouth. The incubation time of COVID-19 coronavirus, which is the period between contamination and the onset of the first symptoms, ranges generally from three to five days [6,7]. However, in some cases it may extend to fourteen days [8,9,10]. During this period, a person may be contagious: they may carry the virus before the appearance of the first symptoms.

Like other known viruses and disease, scientists and researchers have identified some symptoms that may describe a person as infected with a coronavirus. The most common symptoms of COVID-19 are: fever, dry cough, and fatigue [11,12,13,14,15]. Other less common symptoms, like wheezing and pain, nasal congestion, headache, arthritis, sore throat, diarrhea, loss of taste or smell and rash or discoloration of the fingers or toes, may be observed in some cases [16,17,18]. These symptoms, appearing gradually, are usually mild. However, some patients experience very cautious symptoms [19]. As we can see, scientists are still unable to declare exact and definitive symptoms for detection of the coronavirus.

Considering all these factors related to the difficulty in the diagnosis of the coronavirus and the difficulty in its treatment due to its contactless spread in the patient’s environment, it is highly desirable to devise a solution for detecting the presence of coronavirus in a patient using non-invasive and remote methods with minimum involvement of the medical staff. Current solutions focus on taking blood samples or patient’s saliva using a swab test or using X-rays of the lungs [20]. Because of the immediate risk of infections, it is highly desirable to develop contactless, remote solutions for coronavirus detection. One of the solutions is to use cheap medical devices that can be distributed in the community and can be discarded after its use to avoid potential spread of the virus from person-to-person. In the related work, we identify some efforts in this direction, and in the continuity of the previous such efforts, we devise an improved screening mechanism utilizing the latest findings of the disease.

### 1.1. Importance of Remote Screening

While diagnostic tests are used to establish the presence or absence of a disease, the objective of screening is to detect potential indicators. Thus, screening is for high sensitivity, while diagnosis serves for high specificity and demands better accuracy and precision. Screening usually proceeds the diagnosis when there is a large number of potentially at-risk individuals, including those who are asymptomatic. An advantage of screening is that it is inexpensive, simple, and acceptable to both the patients and the medical staff, while diagnosis is usually an expensive and invasive procedure. In many cases, screening is carried out to remove any suspicion of the disease and is often used in combination with other risk factors (diabetes, blood pressure, cardiac complications, FPG levels, etc.) Because screening is cheaper, it can be beneficial to screen a large population that may contain a small number of potential cases. Successful screening may result in identification and successful investigation and treatment of patients at-risk of the disease.

Thus, considering the prevalence of coronavirus disease, rapid screening is highly advantageous, particularly in the areas where advanced medical facilities cannot be found. Due to the shortage of medical staff and lack of administrative resources, a medical device that can be used by the non-experts or that can transmit the results to the medical experts at remote locations is a need of the time.

### 1.2. Detection of COVID-19 Using Wearable Devices

The idea of using sensors for screening or detection of COVID-19 is not new, but many of the existing solutions propose proprietary devices and the focus has been mainly on making ventilators and personal protective equipment (PPE) [21], 3D-printed medical equipment [22], nanotechnology-enabled solutions [23], etc. In the market, many smartphones and wearable devices also offer some sensors for monitoring heartbeat, respiratory rate, sleep quality, etc. and some of them also have API’s for accessing their data in third-party apps. However, due to privacy issues with such devices and given that the proprietary solutions do not provide any flexibility and extensibility, we chose to develop our solution using open-source components. As such the concept of using Free and Open Source scientific and medical Hardware (FOSH) has led to some efforts in combining sensors for the treatment of coronavirus [21,22,23].

In the literature, numerous solutions have been proposed for detection of COVID-19. As time is passing, new information about the disease is constantly appearing, particularly, there have been efforts to make its identification possible with minimum involvement of humans. In the next section, we describe some of these approaches.

### 1.3. Novel Contributions of this Work

Similar to some of the previous approaches, we propose a framework for screening of COVID-19 remotely, with the help of wearable sensorial devices. However, unlike existing approaches, our proposed solution has several novel features. First, the sensors utilized in our approach are cheaper, available off-the-shelf, and can be easily integrated to detect various symptoms, as described above. Each sensor used in our device costs only a few dollars. The advantage is a device can easily be discarded if it is found to be contaminated. Since the sensors are generic, they can be replaced or recomposed into an improved device in the future. Second, the novelty of our approach is that the test can be carried out by experts and non-experts and the results obtained can be inferred by anyone. All the processing is done by the sensors and the accompanying framework. Third, the results can be monitored and analyzed remotely. It means that the wearable device can be used in far areas, while the results can be monitored from professionals in hospitals and clinics at different locations. This is an extremely important aspect of disease detection because as the virus is spread around the globe, governments have limited resources to send medical staff to remote areas. With our proposed solution, the symptoms can be checked on a large population. Fourth, the results are in real-time. It means that a large number of patients can be screened for the illness in a short time. For example, a village of a few hundred people can be tested in a single day with only a few devices. This can be made possible only if the screening can be done in near-real-time. The requirements for such applications imply that the data is kept moving, instead of storing at the source, to process and respond instantaneously, to integrate the stored and streamed data, and to guarantee data safety and availability [24,25]. To meet the near-real-time requirements of the problem, we have identified the sensors that can process and transmit data to the cloud within an upper bound on time, processing, and accuracy. When selecting the sensors, it was ensured that the temperature, heartbeat, cough, and SpO${}_{2}$ sensors were able to process and display the data as soon as a reading was made. The sensors are connected to the Wi-Fi module that is responsible to transmit the received data to the destination without manual intervention, hence, transmission is done in near-real-time. Finally, by using the IoT infrastructure, efficient stream processing and data integration is ensured in the cloud. Our rule-based system for decision-making can evaluate the results in linear-time as compared to the exponential growth in most machine-learning problems [26]. Such algorithms guarantee faster response for any scale of data.

Furthermore, our framework can be helpful in identifying population segments in need of urgent treatment. By analyzing the data of many people in one area, authorities can estimate the severity of the disease and can act urgently on the outcome. Finally, as the procedure involves portable devices, it is easy to transport them from place to place, easily.

The distinguishing features of our framework are that unlike existing approaches that rely on blood or saliva sampling, we use wearable medical sensors that can read and send physiological data of a person to the processing unit of our framework, which can then evaluate the person’s condition based on a number of rules that have been acquired from existing research findings. In essence, the rules are code guidelines extracted from experts with the objective of replacing an expert or reducing the intervention of an expert in medical decision-making. These rules allow to classify a person as healthy or having a possibility of infection by the coronavirus. The advantage of using rules for patient’s classification is that the rules can be updated and evolved with dynamic knowledge from the integration of new clinical guidelines as new findings emerge from time to time. This is important for a disease like COVID-19 because new strains of the virus appear from time to time making it a challenge to have a definite or one-size-fits-all vaccine for its treatment.

The remainder of this manuscript is organized as follows. In Section 2, we provide the necessary background for screening and detection of COVID-19 in general and identify sme related work. Then, in Section 3, we present our framework, the device, the rules and the procedure for screening of COVID-19 patients. Section 4 is dedicated to the description of hardware and software components used by our medical device. We end the paper with a closing discussion in Section 5 followed by the conclusions and ongoing work for improvement of this research.

## 2. Background and Related Work

Before explaining the approaches for screening and detection of COVID-19, we differentiate between this virus and the influenza virus.

### 2.1. The Difference between the Symptoms of the New Coronavirus (SARS-CoV-2) and Influenza

In general, the two viruses have similar symptoms such as fever, cough, headache, muscle pain, and fatigue. For instance, when a person suffers from either of the two diseases, he/she has a fever. However, occurrence of fever is rare in the regular flu and strong in a new coronavirus patient and it may be associated with vomiting and diarrhea [27]. Similarly, fatigue and muscle pain happen, but they are somewhat mild in people suffering from influenza, and severe in a COVID-19 patient. The development of the subsequent symptoms is slow over time in a regular influenza patient, while it is quick in a patient infected with the new coronavirus. It is noticed that the person with the new coronavirus does not suffer from a stuffy nose or a runny nose, while this symptom is observed in the influenza patient and fades within a week [27]. Although headache may be a common symptom of the two illnesses, it is simple and rare in case of regular influenza and strong and continuous in the case of COVID-19. The same patterns apply to chills as they are rare for a person with regular influenza, but they are clearly observed in coronavirus patients. However, while sneezing and sore throat are severe in a patient with regular influenza, they are rare in a person suffering from COVID-19.

Coughing is common in the two types of patients. Nevertheless, it is accompanied by sputum in the case of regular influenza, while in a patient with COVID-19, it is sharp without sputum. The person infected with the new coronavirus suffers mainly from severe pain. The latter is mild and rare in a patient suffering from regular influenza [27]. Chest pain and a feeling of heaviness are common symptoms of the two diseases. In fact, they are mild to moderate in a flu patient, while they are severe and strong in a COVID-19 patient. Flu symptoms and severity can vary depending on the patient’s age and health. The main symptoms are sudden fever varying between 39 ${}^{\circ }$C and 40 ${}^{\circ }$C (102 ${}^{\circ }$F and 104 ${}^{\circ }$F), sudden cough, sore throat, muscle or joint pain, extreme fatigue, and headache [28]. Symptoms, such as nausea, vomiting, diarrhea and stomach pain, may also appear. These symptoms are more common in children. Older adults can feel weak and sometimes be confused without other symptoms.

Thus, while many symptoms are common between influenza and coronavirus, we can use a small subset of the symptoms whose presence can suggest a high probability of developing coronavirus disease in a person. Based on the literature study, our hypothesis is that detection of those few symptoms can be used as in the rapid screening of coronavirus.

### 2.2. Techniques Currently Used for the Detection of COVID-19

One type of COVID-19 detection technique involves the chemical-analysis-based techniques as indicated by Singh et al. [29]. They can be divided into two main classes namely polymerase chain-reaction (PCR)-based techniques:(Real-time PCR [30,31], TaqMan probe-based Real-Time PCR [32], and Droplet Digital-PCR [33,34]), and non-PCR-based techniques (e.g., nucleic acid sequence based amplification, and real-time quantitative loop-mediated isothermal amplification of DNA [35]). These methods are invasive, require specialized laboratory facilities, and, thus, cannot be carried out everywhere.

### 2.3. Using Rules for COVID-19 Detection

Several approaches have used rule-based analysis for detection of COVID-19. Banjar et al. [36] developed an expert system that uses computerized clinical guidelines as rules for COVID-19 diagnosis and management. Salman and Abu-Naser [37] developed a rule-based system using the CLIPS and Delphi languages. The common approach considered in these methods is to codify the expert’s knowledge in the form of rules and evaluate them with respect to the actual conditions present at the time of diagnosis. There have been some cases of using rule-based systems for social distancing and clinical diagnosis as well but they have built in a different context than our study.

### 2.4. Other Proposed Alternatives (Under Investigation) for the Detection of COVID-19

Recently, the authors of [38] aimed to cover COVID-19 related research initiatives and new advances in the use of IoT in smart healthcare techniques. In [39], the authors provided a summary of BioMeTs (Biometric Monitoring Technologies) available for collecting vital signs (blood pressure, heart rate, temperature, respiratory rate, and oxygen saturation) and discussed the strengths and weaknesses of continuous monitoring processes in the coronavirus era.

Next, we attempt to cover some of new proposed techniques in the literature that use sensors for detecting COVID-19.
Use of electrochemical sensors [40]: Traditionally, respiratory infections have been identified by a range of methodologies [41] such as staining, direct fluorescence antibody, etc. Such techniques require costly chemicals and materials, time-consuming preparation of samples, and skilled staff. To tackle these disadvantages, methods like surface plasmon resonance [42], interferometry [43], and field effect transistor [44] were adopted for virus detection. All these methods depend on specialized devices.Use of Smartphone Sensors A new mechanism was proposed for detecting COVID-19 using smartphone sensors in [45]. The proposal offers a cheaper solution, as most radiologists already have smart phones available for various everyday purposes. Not only this, but normal individuals can use the system for virus detection purposes on their phones.Use of Smart Thermometers: In [46], the authors compared smart thermometers and mobile device data to regional influenza and “influenza-like illness” (ILI) monitoring. Similarly in [47], a group of researchers proposed a methodology to identify anomalously high levels of ILI in real-time, at the scale of US counties. Using data from a geospatial network of thermometers involving more than one million users across the US, they identified anomalies by producing precise, county-specific predictions of seasonal ILI from a point before a possible outbreak. Anomalies are strongly correlated with COVID-19 case counts and could provide an early-warning mechanism for locating the epicenters of future possible outbreaks.Wearable Medical Sensors (WMS): A WMS based solution called EasyBand [48] has recently been proposed to restrict the growth of new positive cases by tracking auto-contact and supporting critical social distancing. In an other recent work [49,50], the authors proposed a solution called CovidDeep which uses commercial WMSs for the detection of the COVID-19 virus. Similarly, the authors of [51] developed an application that gathers self-reported symptoms as well as smartwatch and activity tracker data in order to differentiate between COVID-19 negative and positive cases in symptomatic persons.Use of Cough Recognition Techniques: Cough [52] is a characteristic of varied respiratory infections from a common cold to the latest coronavirus infection. Not only does cough exist in humans, but it has been equally found to exist in many species [53]. In the work presented in [54], the authors presented a new technique which detects coughs using a “K-band continuous-wave Doppler radar”. Similarly in [55], a group of scientists have developed an AI model which detects the COVID-19 virus from a forced cough.Use of Arduino and IoT: Magesh et al. [56] used sensors to monitor the temperature and respiratory rate of the COVID-19 cases to develop the mathematical model called the epidemic Susceptible, Infected and Recovered (SIR) to classify the COVID-19 cases in one of the three SIR categories. However, as we describe earlier, temperature and respiratory rates are not sufficient to detect COVID-19 cases. On the same pattern, Al-Shalabi used the temperature sensor to detect COVID-19 [57], which is not an accurate and reliable solution. Ref. [58] proposed an IoT-based solution aiming to increase COVID-19 indoor safety by analysing contactless temperature sensing, mask detection, social distancing check. The temperature sensing relied on Arduino using an infrared sensor or a thermal camera, while mask detection and social distancing checks were performed by leveraging computer vision techniques. The solution could only be helpful in prevention of COVID-19 but could not support COVID-19 diagnosis.

## 3. IoT Framework for Remote Screening of COVID-19

In the absence of any medical tests, not all patients can be followed-up with the traditional diagnosis methods. In the case of coronavirus, this is an issue when the patients should not be in close contact with the caregivers, family members, or doctors. So, it is essential to adopt an innovative technology that facilitates this task. The Internet of Things, especially the Internet of Connected Medical Things (IoMT), is the best technology used to remotely control people affected by the COVID-19 epidemic [59].

Figure 1 shows the framework of our proposed system. The figure explains how IoT can find solutions for problems that cannot be solved using classical techniques utilized in the field of medicine. The realization of the framework is provided by the data providers, resource providers and the support providers. The IoT infrastructure communicates with the three types of providers using different secure communication channels. The data providers are the sensing devices that obtain real-time data from persons and submit it for processing and analysis. The resource providers are the computing and communicating devices that are connected to the infrastructure providing the ability to analyse and visualize the data and facilitate decision-making. Finally, the support providers consist of the network of caregivers and medical facilities that are responsible for patients’ treatment and safety. It is the support network of the person that follows-up once he/she is diagnosed with COVID-19. The support providers are not an active part of the framework but are the users who utilize the framework.

### 3.1. The COVID-19 Screening Device

Our screening device allows physicians or patient-supervising professionals to take physiological measurements and remotely analyze their patients, always know their health conditions, and determine the necessary medical characteristics without any physical and direct contact with them. The device is accompanied by a smartphone application to remotely follow and determine the patient’s health condition if he/she is infected with influenza or coronavirus in a combination of the data from the sensors, as shown in Figure 2. Using patterns from visualization concepts, we use different widgets and different colors to display the results of the sensors as well as the diagnosis, based on data from the sensors. The application uses an algorithm to decide on the status of the patient as without any symptoms or having mild, moderate, or severe symptoms along with individual sensor reading for explanation and evaluation.

Figure 2 shows the prototype implementation of the COVID-19 screening device. The device contains medical sensors connected with a processor and a Wi-Fi module for data processing and transfer to the cloud. The device has two interconnected parts: one placed around the arm, while the other one attached to the frontal part closer to the neck so that the cough intensity and frequency can be determined (see Figure 2b,c). Its purpose is to identify the symptoms of the coronavirus by measuring the temperature, oxygen level in the blood, the heartbeat rate, and determining the severity of the patient’s cough. The on-board process is programmed using Arduino to combine the data and send it to a cloud-storage platform using the Wi-Fi module.

### 3.2. The Rule-Based Analysis of COVID-19

An important part of our framework is the rule-based system for decision-making. While there have been many approaches that utilize machine learning or neural networks for prediction of diabetes, they work on the availability of a large set of data and then identifying patterns in the data for classification or prediction tasks. These approaches cannot be used if very limited data is available or there is no data at all. In such cases, rule-based approaches are a preferred way to perform classification tasks. The basic idea in a rule-based system is to have a rule-base that contains a set of rules. These rules have been learnt from the domain experts or adopted from clinical guidelines and research findings. In the simplest case, rules work on the principles of matching various conditions of the symptoms of a person with the existing knowledge in the rule-base. The rules in our case relate to the absence or presence of a symptom or the range of sensor value above or below a certain threshold value.

For rules definition, we consulted two experts specialized in infectious disease. The consulted doctors identified that SpO${}_{2}$ measurement is a key and essential determinant of COVID-19. If the SpO${}_{2}$ is > 95% with a normal temperature, the patient does not present any sign of disease and the test will be negative and the patient should not go to the COVID-19 center. However, if the SpO${}_{2}$ is between 93% and 94% with enough high temperature (>38), then it is important to get one tested for COVID-19. In addition, we also extracted some rules from the existing literature as discussed in Section 2.

This allowed us to define four classes of screening results. Each class meets a specific set of rules. A patient is evaluated against the rules and is assigned a class based on the conditions stated in each class. These classes as defined as below:Class 0: Non-symptomatic
-SpO${}_{2}$
≥95%;-Cough Rate: NIL;-Heartbeat Rate ≤100 bpm;-Temperature ≤ 37.2 ${}^{\circ }$C;-No headache and pains.-No comorbidities.Class 1: Mild symptoms
-SpO${}_{2}$≥95%;-Cough Rate ≤5/min;-Heartbeat Rate ≤100 bpm;-36 ${}^{\circ }$C ≤ Temperature ≤ 38 ${}^{\circ }$C;-No shortness of breath.-No comorbiditiesClass 2: Moderate clinical symptoms
-93%≤ SpO${}_{2}$
≤94%;-5/min ≤ Cough Rate <30/min;-Heartbeat Rate > 100 bpm;-Temperature ≥ 38 ${}^{\circ }$C.Class 3: Serious clinical symptoms
-SpO${}_{2}$
≤92%;-Cough Rate ≥ 30/min;-Heartbeat Rate >120 bpm;-Temperature > 38 ${}^{\circ }$C.-Occurrence of comorbidities.

While the sensors are useful for detection of vital signs, we have also additional parameters of shortness of breath, headache, and occurrence of any comorbidity (diabetes, heart disease, hypertension, etc.) in our rules. At the moment these parameters are assessed from visual inspection and through question-answering. In the current version, and for screening purposes, it is sufficient to have the confirmation from the patient. In the diagnosis stage, further devices can be used to determine these symptoms. For example, the expert or physician can carry out measurement of glucose level, blood pressure, or performing an ECG for a conclusive outcome. This will only be needed in the case of serious clinical symptoms (class 3). For screening purposes, verbal confirmation of a patient may require several additional questions depending upon the regional guidelines [60] for COVID-19 screening.

### 3.3. Real-Time Screening: Analysis and Visualization

We illustrate the working of our prototype medical device using the following scenario.

A person is suspected of COVID-19 and needs to be screened. The person is instructed to obtain a device from a designated place and wear it according to the directions given by a health worker or professional through some distance. Once a person feels comfortable in wearing the device, it is activated through a mobile interface that starts reading through all the sensors. As each sensor completes its reading, the data is immediately sent to the IoT platform through the Wi-Fi module of Arduino. The data is sent in streams. To be able to detect the cough, the patient is required to wear the device for at least two minutes while the data is being streamed and stored at the cloud, and subsequently, visualized at the dashboard. As soon as all the sensors finish collecting the data, a signal is sent to the mobile device indicating to the health worker to stop the measurement process. A final snapshot is also shown on the mobile screen as it receives data from the IoTplatform. The patient is instructed to remove the device and put it back at a designated place. In case, the patient is screened to be positive, the device may be discarded altogether.

For IoT solutions, we have used the Ubidots [61] IoT platform that is designed with the objective of rapid development of IoT-based solutions. The platform supports streaming of data from sensors and mobile devices, which is then analyzed in real-time. A variety of visualization techniques are available to visualize the data in real-time. Figure 3 shows the interface of our system displaying the data obtained from a patient’s device. This interface is important to consolidate the data obtained from a number of patients. Through advanced analytics, we will be able to carry out statistical analysis, identifying the need for enforcing special precautions in an area infected with coronavirus, etc. However, these features will be integrated in the near future. In addition, we have also developed a dedicated smartphone interface, that is locally connected to the medical device and displays the data in real-time as shown in Figure 4.

## 4. The Hardware and Software Architectural Components

To support openness of architectural components, extensibility, and interoperability, we have chosen the Arduino platform [62]. Arduino is an open-source electronics platform that has been created with the purpose of employing easy-to-use, off-the-shelf electronic components in various software/hardware projects. Together with the Adruino’s Integrated Development Environment (IDE), and the thousands of commercially-available hardware/software components, it helps in the quick prototyping of projects like ours. To include the IoT capabilities in our work, we have used the Ubidots IoT platform, as discussed in Section 4.2.

### 4.1. The Hardware Components

Our approach relies on the use of a combination of sensors that can monitor the physiological signs. Thanks to advancement in medical technology, a number of solutions are available for the given purpose. Our objective was to choose those sensors which are cheaper and can be connected with other hardware and software components, i.e., provide maximum interoperability. We also evaluated the durability, compactness, reliability, certifications, connectivity, availability of developers’ kit, and power-saving performance of sensors. Thus, after a careful analysis and comparison of many available solutions, we chose the sensors as follows.

Table 1 summarizes the hardware components used in the medical device. The microchip chosen for our project is Espressif ESP8266 [63]. The microchip has the ability to connect multiple things in a Wi-Fi network. This is an essential requirement in our case as we are connecting multiple sensors in our solution. The MAX30100 sensor [64] is used as a pulse oximeter and heart rate monitor. It is made up of: two LEDs, a photo Detector, enhanced optics, and low-noise analog signal processing to detect oximetry and heartbeat signals. The sensor MAX30100 operates from 1.8 V and 3.3 V power supplies and can be turned on and off using common software, allowing the power supply to be connected at any time. For heartbeat detection, one LED emits a red light. For pulse rate, only the infrared light is needed by the other LED. This is because the oxygenated blood absorbs more infrared light and passes more red light while deoxygenated blood absorbs red light and passes more infrared light [65]. Both the red light and infrared light are used to measure oxygen levels in the blood. It can be integrated and used efficiently in mobile devices, fitness aids and medical monitoring devices. This sensor also solves the problem of persons with heart disease and those who are in COVID-19 critical situation, as it can be worn on the arm using the electrical pads. This arrangement is very critical for our purpose.

The MAX30205 temperature sensor [66] is able to accurately measure the temperature and provides an alert, overheating, and shutdown output. This unit converts temperature measurements to digital form using a high-precision analog-to-digital sigma-delta converter. The accuracy meets the specifications of the ASTM E1112 thermometer when soldered to the final PCB board. The communication takes place via two-wire serial interfaces in “i2c compatible” mode.

For detection of cough, there are a few approaches that utilize sensors for its detection [52,54,55]. Our supported sensor, the SW-420 is Doppler radar for cough detection. This Continuous-wave (CW) radar uses a voltage-controlled oscillator to continuously transmit a signal. The receiver is always on to detect the echo signal. The CW radar is a simple radar and is easier to integrate into mobile devices. It recognizes the amplitude of the vibration to which it is exposed. Thus, in essence, the vibrations generated from cough are translated into detecting cough using a threshold as shown in Figure 5. The exact threshold at which to identify and separate between different severity of coughs requires data from subjects with varying levels of illness and medical conditions in addition to gender and age differences among different persons.

### 4.2. The Software Components

Table 2 summarizes the main software components of our system. Google Firebase application development software that facilitates the advanced and extensible software creation process by allowing developers to develop iOS, Android and Web apps. In other words, it enables cross-platform, rapid development of mobile and web applications. It is essentially employed to prevent professionals and individuals from participating in the complex process of creating and maintaining a server architecture. In addition, the platform can be run by multiple users at the same time without experiencing any errors. Its intuitive features make it desirable for use in our project.

The Ubidots platform [67] enables the development of IoT applications for manufacturers and individuals in the fields of health, agriculture, smart cities, etc. It is equipped with many features that allow, for example, data to be collected from sensors and visualized via the dashboard. We can access the archives of production data in real-time over a period of 2 years. In addition, it allows configuring conditional events and alerts and activating them via SMS, email, etc. Such alerts and notifications are essential and useful for real-time detection of events, e.g., simultaneous detection of several COVID-19 cases in a proximity will generate an alarm to indicate the severity of infection in that area. However, in this prototype implementation, we have not integrated this option yet.

Combined with APIs that can be accessed via HTTP/MQTT/TCP/UDP protocols, Ubidots provides a simple, secure connection to send and retrieve data to and from the cloud service in real-time. Developers can also combine their own HTML/JavaScript code to customize the data display interface. Because of its security features, extension capabilities, and a wide-range of dashboards for monitoring of real-time sensorial data, we chose it as a technology for our IoT platform.

An important aspect of Ubidots is its focus on the security of communication. All the communication taking place between different components is secured: HTTP with SSL encryption and MQTT with TLS encryption. In addition, Ubidots use token-based secure authorization [68,69]. As the sensors do not provide any data storage and the sensors’ readings are directly sent to the cloud, user’s data is not exposed. Also, in the current implementation, we do not require any personal data except the data sensed by the sensors, but in real-world, the sensor data is augmented by personal information and this should be treated with the same privacy-aware protocols as traditionally carried out by the healthcare institutions. The data protection and integrity at storage location is managed through cloud service providers.

The Arduino Integrated Development Environment (IDE) contains a text-editor for writing code, a message box, a text console, a toolbar with buttons for common functions, and a series of menus. It is connected to Arduino (or the compatible Genuino) devices to download and communicate with programs. The Arduino IDE feature allows: editing a program and compiling this program in the Arduino’s machine language, uploading the program to the Arduino memory, and connecting to the Arduino board via the terminal.

Android Studio is the largest and most popular programming environment in the field of programming and developing mobile applications that run on the Android operating system. It is mainly characterized by easy application development and flexibility. It makes it possible to display the developed applications on a group of different (Android) simulated screens during the programming for a specific application without the need for a set of devices as they support a simulated environment in order to save time and effort.

## 5. Discussion

Point-of-care devices have been one widely-used way for detection of COVID-19. Such devices can take about 5–10 min to produce results [16]. However, the downside of these approaches is that they require some samples (e.g., blood or saliva) from the patient. Some techniques rely on X-ray [20] or CT scan images [70], a process which require specialized devices, a controlled environment, and much longer time than mere a few minutes. On the contrary, we suggest a non-invasive technique that does not rely on patients’ samples but only uses external physiological symptoms of the patients in real-time.

Some approaches also use external sensors as proposed in this work. However, as we have seen in the existing work, those solutions do not fully cover more than a couple of the aspects of detecting COVID-19. For example, while [45] uses “a large number of sensors including cameras, microphone, temperature sensor, inertial sensors, proximity, colour-sensor, humidity-sensor, and wireless chipsets/sensors”, the final decision made by their proposed AI-based system is dependent on chest CT scan images and blood test results. Thus, the contribution of non-invasive, wearable sensors cannot be established.

To the best of our knowledge, our solution is the first to explore the use of temperature, cough, heartbeat, and SpO${}_{2}$ sensors simultaneously to consider the various symptoms which may occur together in a COVID-19 infected patient. The sensors used are very accurate in detecting temperature, heart rate or SpO${}_{2}$, and the vibration level, but the accuracy of the sensor used for detecting cough is dependent on the algorithm for detection of cough because we do not measure cough directly. As our idea is to measure vibrations obtained through cough, the thresholds have been fixed after testing by only a few subjects. In case of some determined frequency of vibration we can say the subject suffers from cough. However, this threshold is suspected to change depending upon further testing that may involve people of both genders from various age groups with different medical backgrounds. In the future work, we will improve our methodology by combining our proposed solution with existing machine learning based techniques for cough detection that will apply a threshold based on various parameters or in combination with other sensors. For the screening purposes, in our current work, it can detect cough reliably from non-cough situations.

Unlike existing solutions that merely propose an architecture, our solution is a working prototype consisting of sensors, smartphone application and the associated IoT infrastructure. By applying IoT infrastructure, we can not only scale the architecture and operation of our system, but it can also be helpful in advanced analysis by applying machine learning and data mining to the data obtained through our system. An added advantage of using open hardware, like Arduino, is the extensibility of our approach in the future as new symptoms of the disease are discovered and new sensors can be integrated. For example, video cameras can be used to detect the aspects of social distancing [58]. Similarly, thermal cameras can be integrated to detect the suspected persons among a group of people or in a crowd. All this can be integrated into the existing solution without changing the previously integrated components. Currently, we are trying to establish the accuracy and reliability of our work by adopting model-based testing techniques [71] in order to validate the proposed approach mathematically.

## 6. Conclusions

The COVID-19 virus has been around for almost a year now and the medical community, scientists, and researchers are trying their best to identify a cure for the disease. At the same time, people around the world are facing issues in determining the state of an individual as healthy or affected by the virus. The state-of-the-art solutions require visiting a hospital or a healthcare facility to perform COVID-19 diagnosis. Considering the numerous difficulties and the associated dangers in its diagnosis, it is preferable to be able to perform the disease detection using wearable devices.

This article proposed a framework for remote screening of the virus using standards-based practice identified in the literature. The framework utilizes sensors combined in the form of a wearable device that can be worn by any individual to know in a few seconds whether the person is healthy or is doubtful of carrying the disease.

The framework requires testing on a large population and at the same time the data obtained through testing can be used for advanced analytics such as outbreak prediction and prevention, population segmenting, as well as helping the government and decision-makers to take appropriate measures. As a future work, and due to unavailability of the required data, we will test our device using data in order to establish its performance.

## Figures and Tables

**Figure 1 ijerph-18-04022-f001:**
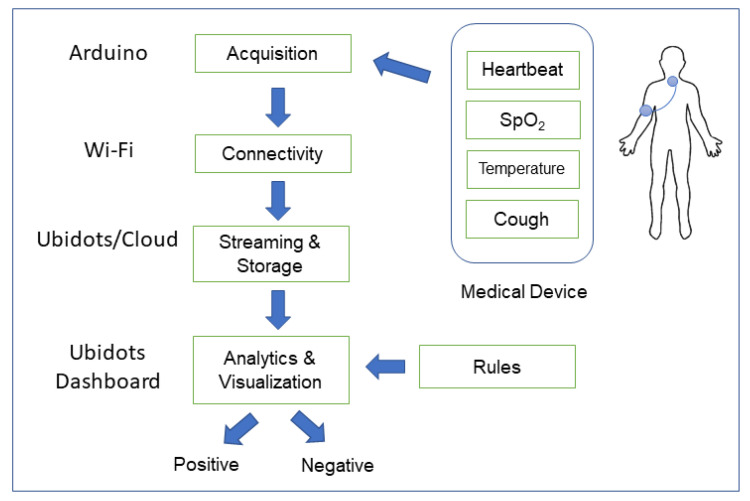
The framework describing the essential features of our approach.

**Figure 2 ijerph-18-04022-f002:**
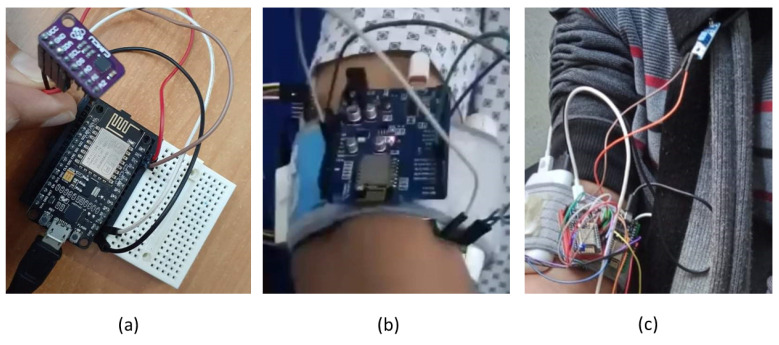
The sensors and the final device in our prototype implementation: (**a**) connecting temperature sensor with Wi-Fi module (**b**) sensor band on the arm (**c**) the wearable configuration of all sensors.

**Figure 3 ijerph-18-04022-f003:**
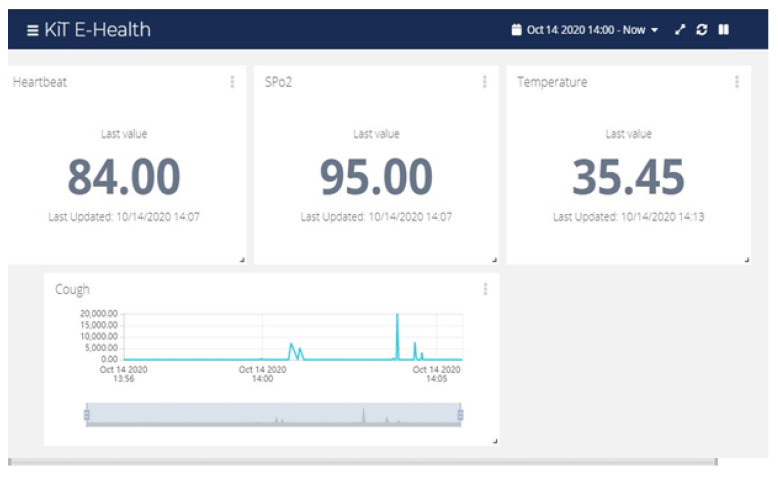
Real-time patient test for heartbeat, SpO${}_{2}$, temperature and cough.

**Figure 4 ijerph-18-04022-f004:**
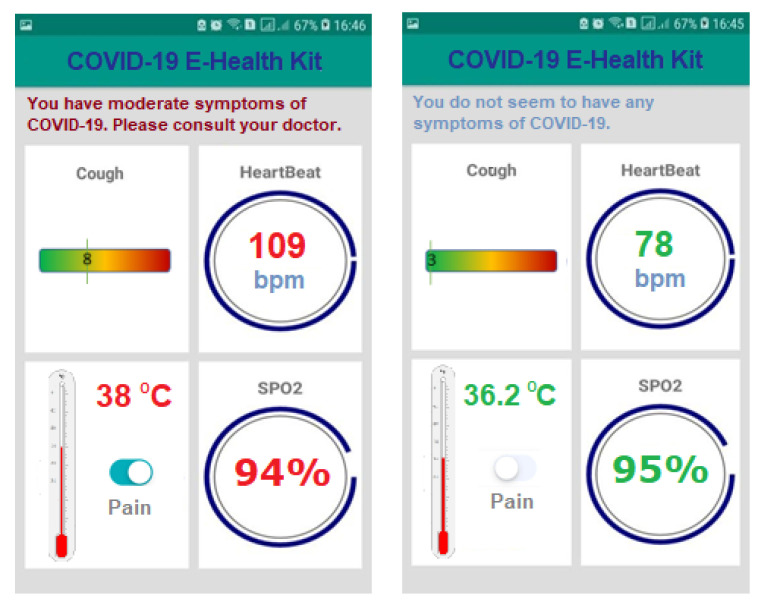
COVID-19 test with Android application for two persons. Data is obtained for sensors and shown separately in its panel. The overall assessment for the patient is also shown.

**Figure 5 ijerph-18-04022-f005:**
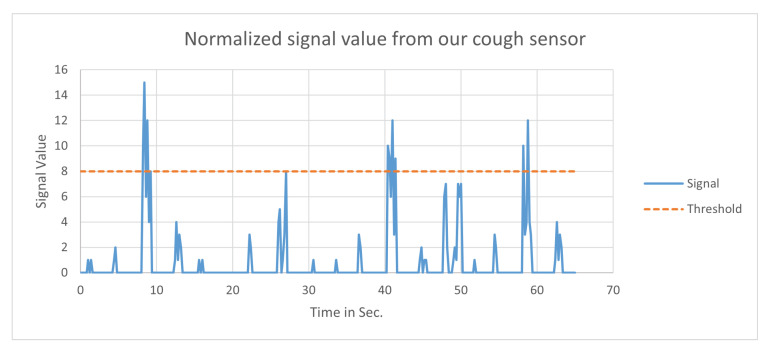
The normalized signal graph obtained from the cough detection sensor.

**Table 1 ijerph-18-04022-t001:** The hardware components and their characteristics.

Purpose	Sensor	Technology	Composition	Performance/Calibration
Measuring human body temperature	MAX30205	Converts the temperature measurements to digital form using a high-resolution, sigma-delta, analog-to-digital converter (ADC)	USB-to-$I^{2}C$ controller along with display units	Meets clinical thermometry specification of the ASTM E1112 (0.1 ${}^{\circ }$C)
Cough detection and variation	SW-420	Doppler radar, continuous-wave (CW) radar, vibration detection	Breakout board that includes comparator LM393	Adjustable on-board potentiometer for sensitivity threshold selection
Pulse/heart-rate	MAX30100	Uses red and infrared frequency of light to determine the percentage of hemoglobin in the blood	Two LEDs, a photo detector, enhanced optics, and low-noise analog signal processing	Programmable from 200 μs to 1.6 ms to optimize measurement accuracy
Wi-Fi connectivity	ESP8266	Integrated TR switch, PLL, regulators, 32-bit CPU	Full TCP/IP stack and microcontroller capability	Wake up and transmit packets in <2 ms

**Table 2 ijerph-18-04022-t002:** The software components used in our prototype implementation.

Software Application	Objective	Usage	Characteristics
Google Firebase	Application creation	For creating client-server architecture	Cross-platform rapid development
Ubidots	IoT data analytics and visualization	To analyse and visualize data from mobile and other computing devices with support for device, app, and resource organization in IoT and cloud infrastructure	Encryption, secure authorization, privacy-aware protocols
Arduino IDE	Sensors connectivity	For programming and customizing the sensors used in the project	Open-source, easy-to-use hardware and software
Android Studio	Android app development	For developing Android-based application interface (Figure 4) and connectivity with the server	Unified environment, structured code modules

## Data Availability

The source code used for device communication (Arduino) can be shared upon request.

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
