# Peer review of "An IoT Framework for Screening of COVID-19 Using Real-Time Data from Wearable Sensors"

_ijerph, 2021, doi:10.3390/ijerph18084022_

Round 1

Reviewer 1 Report

Dear authors,

I hope to contribute constructively to your research with the following review.

The manuscript describes a wearable technology used to remotely and in real-time monitor individuals suspected of being infected with COVID-19. A mobile app and an application in the cloud enhance the wearable. It also contains a description of the technologies used to compose the system. Its applicability can benefit the population and health professionals.

The authors should improve their knowledge of wearable devices and the elements that compose them. Below are the comments that justify the statement.

Broad comments

It is necessary to present and discuss the requirements to affirm that the results are in real-time. There must be clarity between conditions for acquiring biometric signals and conditions for viewing results.

There is confusion between embedded systems. The text does not present the advantages that the chosen embedded brings to application.

Datasheets describe pins and technical characteristics of the components used. Explain the sensor purpose, its principle of operation relating to the type of data generated, which problem is solved, among others; makes the approach more appropriate.

There are fallacies in the description of the technologies used. Report only the advantages that technology brings to application.

What type of calibration utilize, and how was it performed? There is confusion about the concept of calibration. The baselines are references for the calibration.

Future works are the focus of the discussion. The methodology is not sufficient to present results. For example, is the accuracy of the sensors suitable for the application?

Scientific and academic rigor required in manuscripts is lacking in several parts of the text.

Specific comments

Line 3: Make it clear that the problem is a late diagnosis.

Line 16: The abstract must contain the keywords.

Line 43 - 45: The authors suggest an order of reasoning and arrange the manuscript in reverse. The inversion is also evident on lines 90 - 91 and 93 - 94.

Line 105 and 111 - 113: Figures are inanimate objects and do not perform actions. It is the authors who must explain using the figure.

Line 114: Explain why the sensors can be called smart.

Line 203 - 204: Present some items used in this work's app (Figure 2) clarifies better than Figure 10.

Line 231, 234, 237, 240, 243, and 248: Were these parameters measured by sensors? Is the measure indirect? How was it measured?

Line 320 - 321: In my view, this statement also refers to this work. It is necessary to discuss it.

Line 321 - 323: This manuscript does not report the cough sensor used.

Figures 5, 7, and 9 do not help the reproducibility of the manuscript.

I hope to review your improved manuscript again because the idea is good.

Sincerely,

Reviewer.

Reviewer 2 Report

The authors have proposed a mechanism to detect COVID-19 patients using several wearable sensors and develop a wearable bracelet for the detection of COVID-19 patients. 

Strengths:

  • This is timely research in midst of the current COVID-19 pandemic. The paper proposes a wearable device to detect COVID-19 patients, but later it shows multiple sensors, rather than a bracelet.
  • The implementation is well-defined with the images shown. But, similar devices for each specific purpose are already available in the market. So, why this proposed mechanism or device is different than prior efforts and existing devices. 
  • Overall, the paper is well-written, however, the word "discrimination" in the introduction does not provide the intentional meaning and should be revised. Also, some sentences starting from the lowers case after the full stop, and there are some errors that need to be revised.

Weaknesses: 

  • Recently, similar research has already proposed wearable IoT architectures for COVID-19, so it is not clear how their approach is novel or innovative.
  • The accuracy of detection based on this device is in question that needs to be studied in more detail. Moreover, how can we evaluate the effectiveness of this device? 
  • Another critical issue to be addressed, how the data collected from the device would be secured and its impact on user data privacy.
  • It is mentioned that the device has tested 5000 patients and consulted two doctors, but it is not defined how these tests were done on 5000 patients. It requires more discussion on how the tests were conducted. Also, the symptoms are different users, so there could be a number of false positives, and also the accuracy of detection is not clear with such devices. No statistical analysis presented.
  • In the architecture, is the data secured while being transmitted through the network. It is not discussed if the communication from the devices is encrypted or not. The paper mentions that it uses HTTP which is not secure. 
  • Ethical use and privacy concerns are a major issue, especially of medical devices that need to be addressed in the article.  

Reviewer 3 Report

The article presents a system that combines concepts of intelligent sensors in an IoT platform for detection of covid-19. The work has its merit due to the urgency of the theme, however, it still presents itself as an initial proposal. The work is well written but has some flaws: the over-description of the electronic devices, the lack of details on the disease detection algorithm, and the lack of tests on patients. The proposal is much more of a technical description than a scientific contribution. For the validation of the tool, it is necessary to have a practical test of the system for patients with covid-19, but this is not done and allocated only as future works. For the publication of this work, I analyze that these tests are necessary to validate the developed system.

Major reviews:

  • A complete image for the entire framework is missing. The Figures from 3 to 9 split the system. It is need a Figure that presents the system on a patient.
  • Is the system prepared for false positives? As mentioned on Section 3, the SpO2 measurement is the key for covid-19 detection. If the sensor corrected calibrated was not, it will produce a false-positive case. Did the authors think a solution for this case?
  • The main information about the system is described on Section 3.2 to identification covid-19 cases. I recommend the authors to provide more details in this section, use more references to base the decisions for each medical parameter. Moreover, a graph solution could be more effective than enumeration of the data.
  • The algorithm should be more explained. It seems to me that the algorithm is based on rules and decisions. The algorithm was built in this way or another method. Further description is needed.
  • I miss an analysis on patients. The article demonstrated that the system is interesting and can be used to identify the disease. However, it is necessary to check the effectiveness of the system. Is it possible to be tested on patients? What is the hit rate? How many false positives and false negatives? What return do patients provide? Before certifying the functioning of the system, these tests are necessary. It is the last line of the conclusion, but, to present this framework, it is need to test and verify its accuracy and precision.

Minor reviews:

Abstract:

  • The authors should provide more information about the device. The contextualization of the problem has more than 10 lines, however, the solution is briefly mentioned.
  •  

Introduction

  • The objective and justification of the research are described on section 2.2. I recommend to the authors insert this information at the end of the introduction because the readers expect this information here, with the detailed of what they will find in the text.

COVID-19 Detection

  • “(…) muscle pain and fatigue”. Insert a comma after pain.
  • The two first paragraphs are well written. To summarize the information, I suggest a table with the information about the two diseases.

IoT Framework for Remote Detection of COVID-19

  • “The following figure (…)”. Please, enumerate the figures and cite as in template: Figure 1.
  • In Figure 1, were all the images made by the authors? If not, I suggest referencing them. In addition, the legend is very short. I recommend to the authors explain more the figure by the caption.
  • “(…) explained later, to (…)”. Please, remove “explained later”.
  • -“ Andruino’s Integrated Development Environment”. I think that the correct is “Arduino’s”.
  • -The Figure from 3 to 9 are extremely large. Please, resize them, leaving the figures with a smaller format so as not to disturb the dynamics of reading of the text.
  • Line 145: the “Pulse Oximeter and (…)” is a title. Please, correct it.
  • Line 149: “the sensor”. Insert the capital letter in “The”.
  • Line 172. Same commentary that Line 145, “MAX30205 (…)” is a subtitle.
  • Line 198. Same commentary that Line 145, “Ubidots IoT (…)” is a subtitle.
  • Line 210. Same commentary that Line 145. “The open-source Arduino(…)” is a subtitle.

Round 2

Reviewer 1 Report

Dear authors,

I am happy to review your manuscript again.

The manuscript describes a wearable technology used to remotely and in real-time monitor individuals suspected of being infected with COVID-19. A mobile app and an application in the cloud enhance the wearable. It also contains a description of the technologies used to compose the system. Its applicability can benefit the population and health professionals.

Broad comments

There are still fallacies in the description of the technologies used. It tends to be an advertisement for such technologies. This part should be more objective, relating to the function of each sensor.

It is necessary to clarify the objective of using the Arduino together with ESP8266. The ESP8266 is a device that can perform the same tasks with efficiency, similar to the Arduino.

The accuracy of the sensors used must be evident. The authors state that one of the sensors does not have adequate accuracy to use in the application. In this way, hardware and software are not reliable.

Specific comments

Line 112-113: There is confusion about the concept of real-time.

I see no need for figures 2a, b, and c; for the same reasons already stated in the previous review. The neck device should show, as it makes up the wearable.

Line 289-313: Class 0 is defined by the parameters measured, together with “No headache and pains.” and “No comorbidities.”. However, the wearable does not measure the last two. Therefore, either enter measurements for such parameters or not use them to define such a class. The same happens for “No shortness of breath.” and “No comorbidities” in class 1 and “Occurence of comorbidities.” in class 3.

Table 1 should contain the accuracy of all sensors.

Line 428-430: This statement is only true together with the accuracy for all sensors. If the cough sensor is not accurate enough, it makes no sense to use it. Therefore, the wearable device does not meet the proposed objective.

Line 443-445: Hardware accuracy related to the references of interest is the method that will bring reliability to this work, considering what has been done so far. I cannot recommend accepting this work until I have solved this problem. I agree that validation may be the subject of future works.

Congratulations on the evolution of the work, but more efforts are needed.
Sincerely,
Reviewer.

Reviewer 3 Report

The authors have done extensive revision work, both in text and in content. The work has a delimited scope, the framework's area of expertise is clear, and its contributions fit as a scientific contribution. Only a few aesthetic aspects are necessary to provide greater fluidity during reading and technical points regarding the template should be attended.

Section 1

Line 35: A hyperlink is presented in the text. I recommend converting it in a reference or additional information.

Section 2

The Figure 1 is presented before mentioned in the text. Please, first mention and, then, present the figure.

Section 3

Line 284: “(…)while the other one attached to the frontal part of the neck”. It is interesting a figure to illustrate this process.

Line 327 : A hyperlink is presented in the text. I recommend the same process previously described.

The Figure 3 is also presented before mentioned in the text. Please, first mention and, then, present the figure.

Section 4

Line 339: Another hyperlink.
